# A Hybrid Method Using HAVOK Analysis and Machine Learning for Predicting Chaotic Time Series

**DOI:** 10.3390/e24030408

**Published:** 2022-03-15

**Authors:** Jinhui Yang, Juan Zhao, Junqiang Song, Jianping Wu, Chengwu Zhao, Hongze Leng

**Affiliations:** College of Meteorology and Oceanography, National University of Defense Technology, Changsha 410000, China; yangjinhui@nudt.edu.cn (J.Y.); junqiang@nudt.edu.cn (J.S.); wjp@nudt.edu.cn (J.W.); zhaochengwu12@nudt.edu.cn (C.Z.); hzleng@nudt.edu.cn (H.L.)

**Keywords:** chaotic time series prediction, Koopman, machine learning, Hankel matrix

## Abstract

The prediction of chaotic time series systems has remained a challenging problem in recent decades. A hybrid method using Hankel Alternative View Of Koopman (HAVOK) analysis and machine learning (HAVOK-ML) is developed to predict chaotic time series. HAVOK-ML simulates the time series by reconstructing a closed linear model so as to achieve the purpose of prediction. It decomposes chaotic dynamics into intermittently forced linear systems by HAVOK analysis and estimates the external intermittently forcing term using machine learning. The prediction performance evaluations confirm that the proposed method has superior forecasting skills compared with existing prediction methods.

## 1. Introduction

A chaotic system refers to a deterministic system where there are irregular movements that appear to be random, and its behavior is uncertain, unrepeatable, and unpredictable. The high sensitivity to the initial conditions and the fact that they are inherently unpredictable are the main characteristics of the chaotic systems. Chaotic phenomena are ubiquitous in several scientific fields, such as atmosphere motions [1], population dynamics [2,3,4], epidemiology [5], and economics. It has been a hot topic in such fields and attracted the attention of many people. It is worth noting that the chaotic system is not completely random as its name suggests but has certain structure and patterns. However, due to the lack of understanding of the dynamic mechanism of chaotic systems, the prediction of chaotic time series is still a very important but challenging problem.

With the development of big data and advanced algorithms in machine learning, it has become a new research direction to solve prediction problems of chaotic systems using a data-driven way. Several empirical models to predict chaotic time series based on machine learning are proposed. Many famous artificial neural networks (ANN) models such as Radial Basis Function (RBF) neural network [6], neuro-fuzzy model with Locally Linear Model Tree (LoLiMoT) [7], feedforward neural network [8], multi-layer perceptron (MLP) [9], recurrent neural networks (RNN) [10], finite impulse response (FIR) neural network [11], deep belief nets (DBN) [12], Elman neural network [11,13,14], and wavelet neural network (WNN) [15,16] have been introduced in the literature.

However, the setting of neural network model parameters will greatly affect the performance of these models. Consequently, a substantial amount of work has also been put into the optimization algorithm and parameter settings. Min Gan et al. present a state-dependent autoregressive (SD-AR) model, which uses a set of locally linear radial basis function networks (LIRBFNs) to approximate its functional coefficients [17]. In addition, Pauline Ong and Zarita Zainuddin presented a modified cuckoo search algorithm (MCSA) to initialize WNN models [16]. A hybrid learning algorithm, called HGAGD, which combines genetic algorithm (GA) with gradient descent (GD) is proposed to optimize the parameters of a quantum-inspired neural network (QNN) [18]. In the proposed methodology, the embedding method is used along with ENN to predict the residual time series [13]. A single hidden Markov model (HMM) combined with fuzzy inference systems is introduced for time series predicting [19]. What is more, many hybrid methods are also developed for improving the performance of these prediction models [20,21].

As mentioned above, model structure and parameter tuning are important factors for chaotic time series prediction with machine learning, and a lot of research has focused on it. To simplify the learning model, a hybrid method using Hankel Alternative View Of Koopman (HAVOK) analysis and machine learning (HAVOK-ML) is developed to predict chaotic time series in this research. Hankel Alternative View Of Koopman (HAVOK) analysis was proposed by Brunton [22]. It combines the delay embedding method [23] and the Koopman theory [24] to decompose chaotic dynamics into a linear model with intermittent forcing. HAVOK-ML decomposes chaotic dynamics into intermittently forced linear systems with HAVOK; then, it estimates the forcing term using machine learning. Essentially, the prediction of the chaotic time series using the HAVOK-ML method is conducted as solving linear ordinary differential equations, which can be calculated efficiently. It can take different types of regression methods such as Linear Regression or Random Forest Regression (RFR) [25] into the prediction framework and combines the advantages of HAVOK theory and machine learning. Therefore, it can obtain better prediction results than directly using those machine learning models.

This paper is organized as follows. Section 2 briefly describes the theory of the HAVOK analysis combined with the machine learning method for time series prediction. Section 3 applies the proposed combined method to perform multi-step ahead prediction for some well-known chaotic time series and also compares the obtained prediction performance with that of existing prediction models. Finally, conclusions are given in Section 4.

## 2. HAVOK-ML Method

Consider a nonlinear system of the form:(1)$$\frac{dx(t)}{dt}=f\left(x\left(t\right)\right)$$
where $x\left(t\right)\in R^{n}$ is the state of the system at time t and f denotes the dynamics of the system. For a given state $x(t_{0})$ at time $t_{0}$, $x(t_{0}+t)$ can be given discretely by:(2)$$x\left(t_{0}+t\right)=x\left(t_{0}\right)+\int_{t_{0}}^{t_{0}+t}f\left(x\left(\tau \right)\right)d\tau$$

Generally speaking, for an observed chaotic time series x(t), the governing equation f is highly nonlinear and unknown. HAVOK analysis [22] provides linear representations for those unknown nonlinear systems. A Hankel matrix H, for a single measurement x(t), by taking singular value decomposition (SVD), is given by:(3)$$H=\left(\begin{array}{cccc} x_{(}t_{1}) & x_{(}t_{2}) & \ldots & x_{(}t_{p})\\ x_{(}t_{2}) & x_{(}t_{3}) & \ldots & x_{(}t_{p}+1)\\ \vdots & \vdots & \ddots & \vdots \\ x_{(}t_{q}) & x_{(}t_{q+1}) & \ldots & x_{(}t_{m}) \end{array}\right)=U\Sigma V^{T}$$
where m=p+q−1, *p* and *q* are two parameters that determine the dimension of H. The columns of H are defined by:(4)$$h\left(i\right)={\left[x\left(t_{i}\right),x\left(t_{i+1}\right),\cdots ,x\left(t_{i+q-1}\right)\right]}^{T}\left(i=1,\cdots ,p\right)$$
then
(5)H=[h(1)h(2)⋯h(p)]

Usually, H can be well approximated by the first *r* columns of U, V. According to the HAVOK analysis [22], the first r−1 variables in V can be built as a linear model with the last variable $v_{r}$ as a forcing term:(6)$$\frac{dv_{r-1}\left(t\right)}{dt}=Av_{r-1}\left(t\right)+B\left(t\right)$$
where $v_{r-1}={\left[v_{1},v_{2},\cdots ,v_{r-1}\right]}^{T}$ is the vector of the first r−1 eigen-time-delay coordinates. Note that Equation (Equation 6) is not a closed model because $v_{r}\left(t\right)$ is an external input forcing. In the linear HAVOK model, matrix A and vector B may be obtained by the Sparse Identification of Nonlinear Dynamics (SINDy) algorithm [26] or by a straightforward linear regression procedure. $v_{r}$ is given by the rth column of V.

A machine learning method is used to predict $v_{r}\left(t+1\right)$ by using previous observed values [x(t−DΔt),x(t−(D−1)Δt,⋯,x(t−Δt)], as shown in Figure 1. Suppose $v_{r}$ evenly varies within interval [t,t+1]. The evolution of $v_{r}\left(t\right)$ can be approximated by:(7)$$\frac{dv_{r}\left(t\right)}{dt}=\left(v_{r}\left(t+1\right)-v_{r}\left(t\right)\right)/\Delta t$$Then, the first r−1 variables $v_{r-1}\left(t+1\right)$ are obtained by solving the linear model Equation (Equation 6):(8)$$H=\left(\begin{array}{c} v_{r-1}\left(t+1\right)\\ v_{r}\left(t+1\right)\\ v_{r}\left(t+1\right)-v_{r}\left(t\right) \end{array}\right)=exp\left(\begin{array}{ccc} Adt & Bdt & 0\\ 0 & 0 & I\\ 0 & 0 & 0 \end{array}\right)\left(\begin{array}{c} v_{r-1}\left(t\right)\\ v_{r}\left(t\right)\\ v_{r}\left(t+1\right)-v_{r}\left(t\right) \end{array}\right)$$Assume that the integration starts at time $t_{p}$ for an input h(p). Then, the next step of h(p+1) can be written as:(9)$$h\left(p+1\right)=U\Sigma v_{r}\left(t_{p}+1\right)$$
where $v_{r}=\left[v_{1},v_{2},\cdots ,v_{r-1},v_{r}\right]$ is a vector containing the first *r* variables. In order to evaluate the efficiency of HAVOK-ML, RMSE, NMSE, and $R^{2}$ score defined below are used as a performance index.
(10)$$\;\;\mathrm{RMSE}=\sqrt{\frac{1}{N}\sum_{i=1}^{N}{\left(y_{i}-{\hat{y}}_{i}\right)}^{2}}$$
(11)$$\mathrm{NMSE}=\frac{\sum_{i=1}^{N}{\left(y_{i}-{\hat{y}}_{i}\right)}^{2}}{\sum_{i=1}^{N}{\left(y_{i}-{\bar{y}}_{i}\right)}^{2}}$$
(12)$$\mathrm{error}=\frac{\sum_{i=1}^{N}{\left({\hat{y}}_{i}-{\bar{y}}_{i}\right)}^{2}}{\sum_{i=1}^{N}{\left(y_{i}-{\bar{y}}_{i}\right)}^{2}}$$
where $y_{i},{\hat{y}}_{i}$, and $\bar{y}$ represent the observed data, the predicted data, and the mean of the observed data, respectively. The Root Mean Squared Error (RMSE) and the Normalized Mean Squared Error (NMSE) are used to assess the accuracy of the prediction and to compare the results with those of the literature. The $R^{2}$ score is used to evaluate the score of the machine learning based prediction for $v_{r}$.

## 3. Numerical Experiments

In this section, three different type of time series—Lorenz [1], Mackey–Glass [26], and Sunspot—are applied to verify our proposed HAVOK_ML method. The parameters adopted in the HAVOK analysis of these series are listed in Table 1.

### 3.1. Lorenz Time Series

The Lorenz system [1] is among the most famous chaotic systems, which is described by:(13)$$\begin{array}{cc} \frac{dx}{dt} & =\sigma (y-x)\\ \frac{dy}{dt} & =x(\rho -z)-y\\ \frac{dz}{dt} & =xy-\beta z \end{array}$$The chaotic time series is obtained with parameters σ=10, ρ=28, β=8/3, and dt=0.01 in the second sampling. In this study, only the time series of variable x(t), shown in Figure 2, is considered.

In this research, HAVOK-ML decomposes chaotic dynamics into intermittently forced linear systems by HAVOK analysis; the settings of HAVOK analysis are given in Table 1, and the sampling time step for each system is consistent with other references listed in Table 2. However, according to the advice in paper [14], the samples of the Lorenz system are interpolated at 0.001 s resolution in the HAVOK analysis.

By using HAVOK analysis for the training data, a linear HAVOK model (Equation (Equation 6)) is developed. As shown in Figure 3, matrix A and vector B are sparse, and the reconstruction of $v_{1}\left(t\right)$ and x(t) is coherent with the actual values for the full range of time. Since the $v_{r}$ (Figure 4) is not smooth enough, many experiment results demonstrate that the RFR method [25] can predict the $v_{r}$ best. Hence, an RFR method is adopted to train and estimate the next step $v_{r}\left(t+1\right)$ based on previously observed values [x(t−40),x(t−35),⋯,x(t−10),x(t−5)]. The samples from the 3rd to the 100th seconds are spilled into training set (first 80%) and test set (20%). The $R^{2}$ score for the RFR method on the test set is 0.87. We can observe that the estimated results are mainly consistent with the actual values, as shown in Figure 4.

In the next experiment, the HAVOK-ML method is used in the N-step recursive prediction of Lorenz time series. In the recursive prediction, the current predicted values are used for next predictions without any correction to the actual values. A comparison between the multi-step predicted values and the original time series, with 1000 testing samples, is shown in Figure 5. It can be seen that in the initial steps of predict (less than 10), the prediction results are coherent with the actual values. The error increases with predict steps, especially at the region near the extreme point of the curve. The RMSE of the prediction function of time is presented in Figure 6. It can be observed that the error quickly increases with the increase of the predicted time, which means that a long-term prediction is basically impossible. At step 10, the obtained RMSE is 9.003× 10${}^{-3}$ (Figure 6), which is significantly better than the result of the literature (0.014) [27].

Table 2 presents the one-step ahead prediction errors (RMSE and NMSE) for the proposed method as well as some results obtained by existing methods, which were extracted from the literature. It can be shown that the RMSE index of the proposed method is optimal, while the NMSE index shows that the proposed method is second only to the functional weight WNN state-dependent AR (FWWNN-AR) model [15].

### 3.2. Mackey–Glass Time Series

The Mackey–Glass chaotic time series has been introduced as a white blood cell production [26]. It is described by:(14)$$\frac{dx(t)}{dt}=\frac{ax(t-\tau )}{1+x^{r}\left(t-\tau \right)}-bx\left(t\right),t>0$$
where a=0.2, b=0.1, r=10 and τ=17, similar to other published papers presented in Table 3. The Mackey–Glass equation is solved using the delay differential equation method dde23 of MATLAB. A chaotic time series samples set of 25,000 lengths, with time step dt=0.1, is generated. The samples from the 300th to 2000th seconds, shown in Figure 7, are chosen as the training set, while the rest is used as the test set.

The HAVOK analysis settings for Mackey–Glass time series are given in Table 1. The rows in the H matrix are for q=5, and the rank of the SVD decomposition is r=5. More details on the HAVOK analysis and the multi-step ahead prediction for the Mackey–Glass time series are presented in Figure A1, Figure A2, Figure A3 and Figure A4 in Appendix A. By considering the properties of the $v_{r}$ curve, an LLN model with the LoliMoT optimization method [7] is determined through experiments as a regressor to predict $v_{r}$. A comparison between the prediction accuracies of the proposed method and other models of the literature are summarized in Table 3. As shown in Table 3, whether RMSE or NMSE, the effect of the proposed method outperforms the existing models.

### 3.3. Sunspot Time Series

The number of sunspots observed in the solar surface varies within a period of approximately 11 years. The variation of the number of sunspots has a large impact on Earth and on the climate. The monthly smoothed sunspot number time series, observed by the Solar Influences Data Analysis Center (http://sidc.oma.be/index.php (accessed on 7 August 2021)), is used for training and forecasting the trend of the sunspot variation. In order to compare the results with those extracted from the existing published papers of the literature (listed in Table 4), the series from November 1834 to June 2001 (2000 data points) are chosen and scaled between 0 and 1. The first 1000 samples are selected as the training set (Figure 8), and the remaining 1000 samples are used as the testing set. The settings of HAVOK analysis for sunspot time series are given in Table 1. Similar to the Mackey–Glass time series, an LLN model is used as the regressor to predict $v_{r}$. The details of the HAVOK decomposition and the multi-step prediction for the sunspot time series are shown in Figure A5, Figure A6, Figure A7 and Figure A8 in Appendix A. A comparison between the prediction errors (RMSE and NMSE) obtained with the 1000 samples of the testing set and with other models are presented in Table 4. It can be seen that the proposed HAVOK-ML method outperforms the existing methods in predicting the sunspot chaotic time series.

## 4. Discussion and Conclusions

In this paper, a HAVOK-ML method combining the HAVOK analysis with machine learning to predict chaotic time series is proposed. Based on the HAVOK analysis, the observed chaotic dynamic system could be reconstructed as a linear model with an external intermittent forcing. A machine learning method was applied to predict the external forcing term by using previously observed values. Finally, the combination of the HAVOK analysis with machine learning produces a closed model for prediction. It is worth noting that the machine learning method used in HAVOK-ML will vary depending on the property of the external forcing term. The developed method has been validated for multi-step ahead prediction of several classic chaotic time series (the Lorenz time series, the Mackey–Glass time series, and the Sunspot time series). The experimental results show that our method can produce accurate forecasts even with simple machine learning algorithms. The prediction performance of the proposed method has been compared with other forecasting models of the literature. The comparison shows that the proposed method outperforms the existing ones in terms of superior forecasting ability. Although HAVOK-ML can be combined with different machine learning methods, it does not give suggestions on how to choose machine learning methods for different time series forecasting problems. This is worth studying in the future.

## Figures and Tables

**Figure 1 entropy-24-00408-f001:**
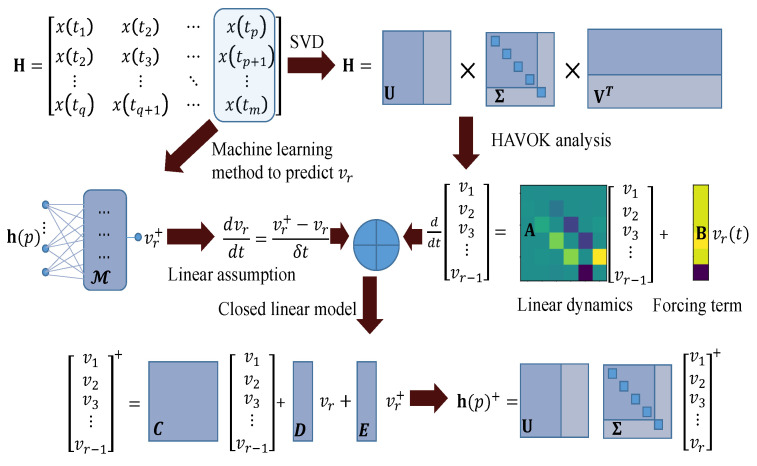
The architecture of the HAVOK-ML method to perform one-step prediction. The SVD of Hankel matrix H yields eigen time series $V^{T}$. On the one hand, the HAVOK analysis gives a linear system for the first r−1 variables with $v_{r}\left(t\right)$ as an external input. On the other hand, by using the machine learning method, the evolution of $v_{r}\left(t\right)$ can be established. Hence, a closed linear model for the first *r* variables is available. The symbols with superscript + stand for values at the next step t+1.

**Figure 2 entropy-24-00408-f002:**
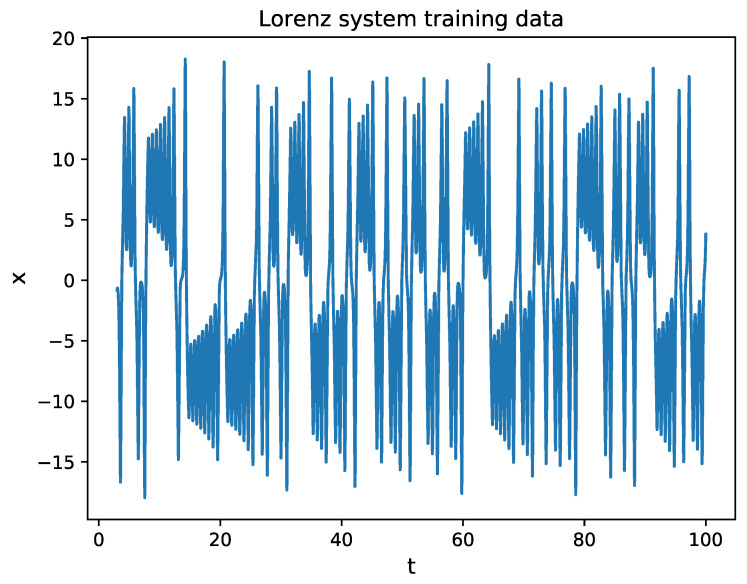
The time series x(t) in the Lorenz system. The initial condition is (−8, 8, 27). The training data are chosen from the 3rd to the 100th seconds.

**Figure 3 entropy-24-00408-f003:**
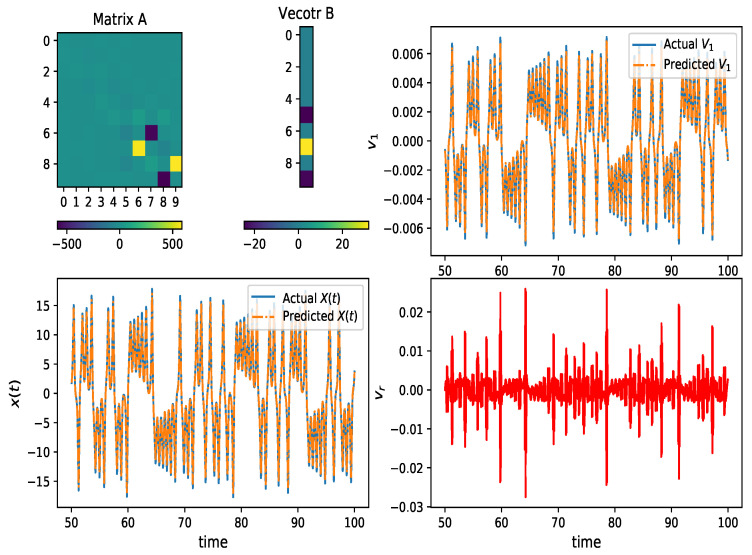
HAVOK analysis for Lorenz chaotic series x(t). From upper-left to bottom-right: matrix A, vector B, reconstruction of $v_{1}\left(t\right)$ using the linear HAVOK model with forcing $v_{r}\left(t\right)$, reconstruction of x(t) and the input of external forcing $v_{r}\left(t\right)$.

**Figure 4 entropy-24-00408-f004:**
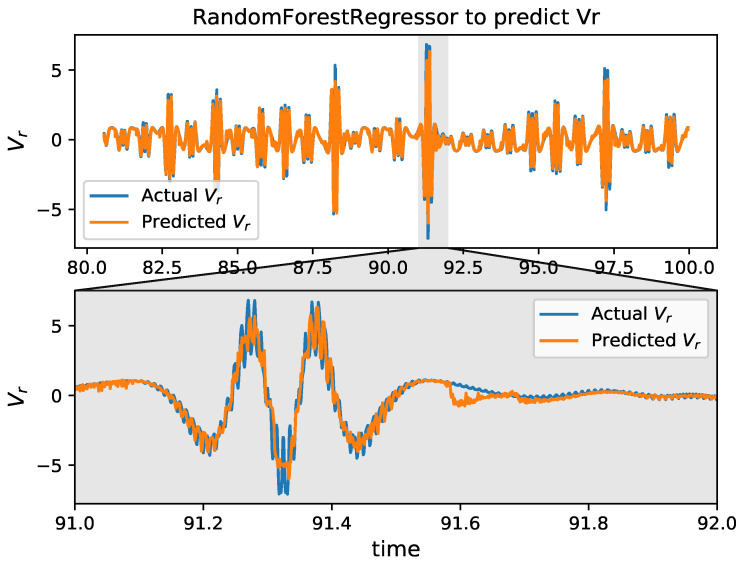
The random forest regressor for $v_{r}$, using previously observed values at [x(t−40),x(t−35),⋯,x(t−10),x(t−5)] to predict the next time value at $v_{r}\left(t+1\right)$, with Δt=0.001.

**Figure 5 entropy-24-00408-f005:**
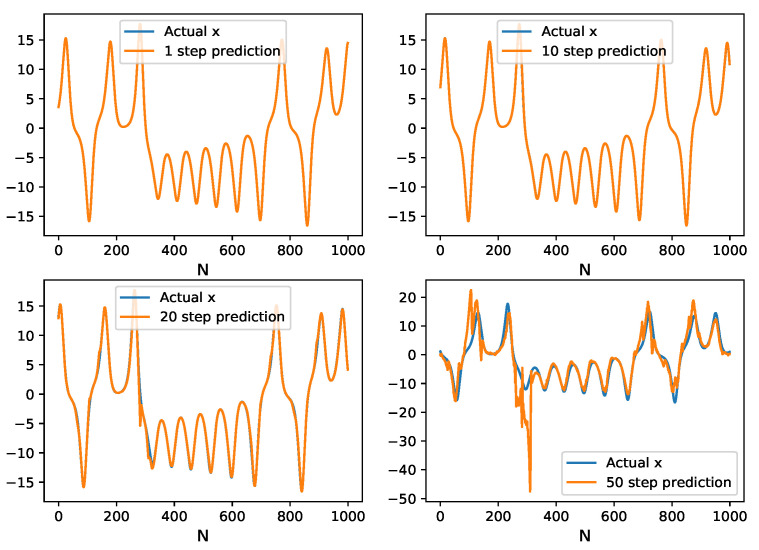
Comparison of the original time series samples and the multi-step predicted values with one-step length of 0.01 s on Lorenz time series.

**Figure 6 entropy-24-00408-f006:**
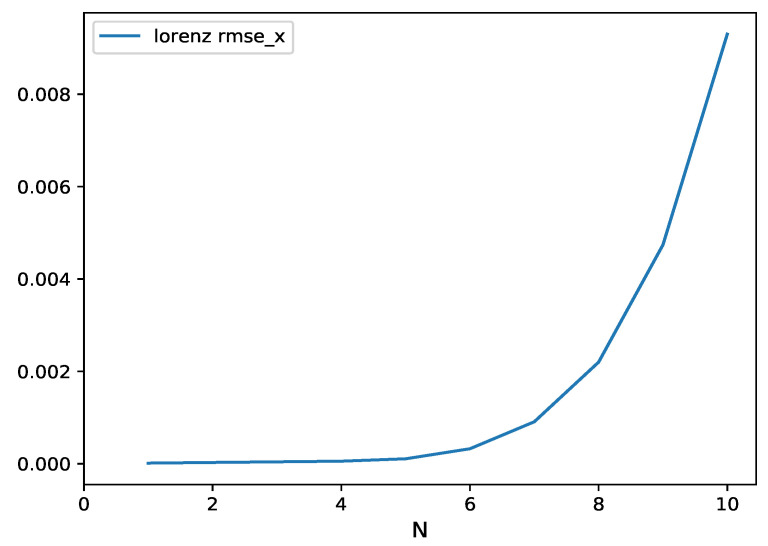
Lorenz time-series RMSE of multi-step ahead prediction, function of the number of steps (N), with one-step length of 0.01 s.

**Figure 7 entropy-24-00408-f007:**
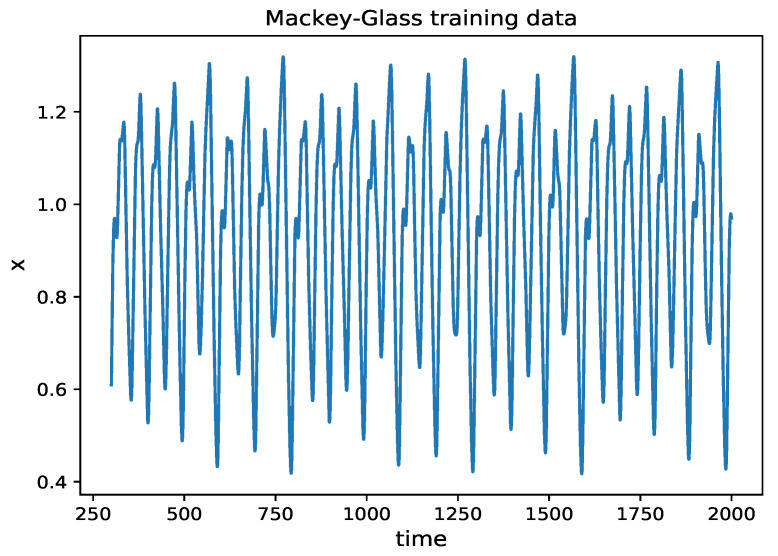
Time series of Mackey–Glass system. The initial condition is 0.8, and the training data are chosen from the 300th to 2000th seconds.

**Figure 8 entropy-24-00408-f008:**
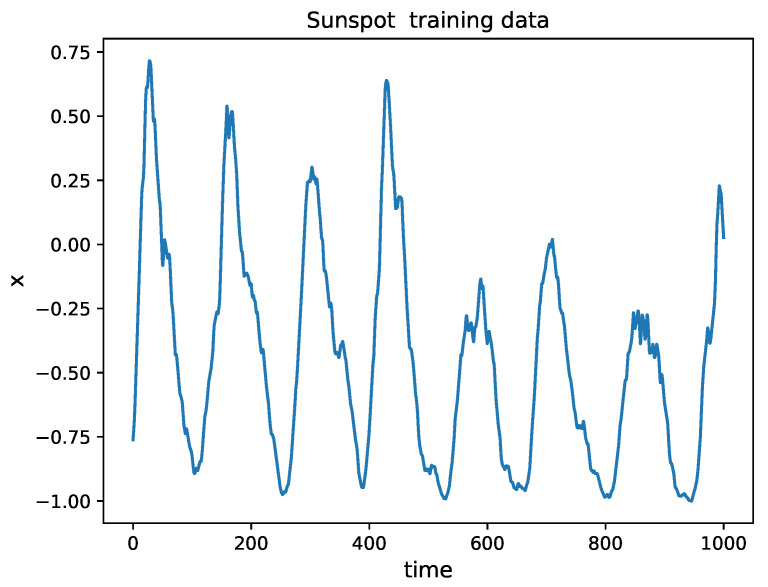
Time series of sunspot normalized to [−1, 1]. The training period ranges between November 1834 and March 1918.

**Table 1 entropy-24-00408-t001:** HAVOK analysis parameters for each system.

System	Samples	dt	Δt	q	Rank (r)	Regressor for $v_{r}$
Lorenz	20,000	0.01 s	0.001 s	40	11	RandomForest
Mackey-Glass	50,000	0.1 s	/	5	5	LoLiMoT
Sunspot	2000	1 month	0.02 month	140	7	LoLiMoT

**Table 2 entropy-24-00408-t002:** Comparison of the models in one-step predicting for Lorenz chaotic series x(t), with 1000 testing samples. The last row represents the proposed HAVOK-ML combined with the RFR method. The highest prediction accuracies achieved by the models are shown in bold.

Model	RMSE	NMSE	Reference
Deep Belief Network	1.02 × 10${}^{-2}$	/	[12]
Elman–NARX neural networks	1.08 × 10 ${}^{-4}$	1.98 × 10${}^{-10}$	[13]
WNN	/	**9.84 × 10${}^{-15}$**	[15]
Fuzzy Inference System	3.1 × 10${}^{-3}$	/	[19]
Local Linear Neural Fuzzy	/	9.80 × 10${}^{-10}$	[7]
Local Linear Radial Basis Function Networks	/	4.53 × 10${}^{-12}$	[27]
WNNs with MCSA	8.20 × 10${}^{-3}$	1.22 × 10${}^{-6}$	[17]
HAVOK_ML(RFR)	**1.43 × 10${}^{-5}$**	3.23 × 10${}^{-12}$	

**Table 3 entropy-24-00408-t003:** Comparison of the models in six-time step ahead predicting Mackey–Glass time series, with 4000 testing samples. The last row shows the proposed HAVOK-ML method with the LLN model as the regressor. The values in bold are the highest prediction accuracies achieved by the models.

Model	RMSE	NMSE	Reference
ARMA with Maximal Overlap Discrete Wavelet Transform	/	5.3373 × 10${}^{-7}$	[16]
Ensembles of Recurrent Neural Network	7.533 × 10${}^{-3}$	8.29 × 10${}^{-4}$	[20]
Quantum-Inspired Neural Network	9.70 × 10${}^{-4}$	/	[28]
Recurrent Neural Network	6.25 × 10${}^{-4}$	/	[18]
Type-1 Fuzzy System	4.8 × 10${}^{-4}$	/	[21]
Fuzzy Inference System	7.1 × 10${}^{-4}$	/	[19]
WNNs with MCSA	5.60 × 10${}^{-5}$	6.25 × 10${}^{-8}$	[17]
HAVOK_ML(RFR)	**9.92 × 10${}^{-6}$**	**1.86 × 10${}^{-9}$**	

**Table 4 entropy-24-00408-t004:** Comparison of the models in one-time step ahead predicting sunspot time series, with 1000 testing samples. The last row shows the proposed HAVOK analysis with the LLN model as the regressor. The values in bold are the highest prediction accuracies achieved by the models.

Model	RMSE	NMSE	Reference
Elman-NARX Neural Networks	1.19 × 10${}^{-2}$	5.90 × 10${}^{-4}$	[13]
Elman Recurrent Neural Networks	5.58 × 10${}^{-2}$	1.92 × 10${}^{-2}$	[29]
Ensembles of Recurrent Neural Network	1.52 × 10${}^{-2}$	9.64 × 10${}^{-4}$	[20]
Fuzzy Inference System	1.18 × 10${}^{-2}$	5.32 × 10${}^{-4}$	[19]
Functional Weights WNNs State Dependent Autoregressive Model	1.12 × 10${}^{-2}$	5.24 × 10${}^{-4}$	[21]
WNNs with MCSA	1.13 × 10${}^{-2}$	5.30 × 10${}^{-4}$	[17]
HAVOK_ML(RFR)	**4.25 × 10${}^{-3}$**	**7.40 × 10${}^{-5}$**	

## Data Availability

The code and the dataset supporting the results of this article are available in the https://gitee.com/yangjinhui11/havok_py/tree/master (accessed on 14 August 2021).

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
