# Peer review of "A Hybrid Method Using HAVOK Analysis and Machine Learning for Predicting Chaotic Time Series"

_entropy, 2022, doi:10.3390/e24030408_

Round 1
Reviewer 1 Report
The paper's novelty lies in combining several methods of decomposition and forecasting chaotic time series (HAVOK-ML). Three well-recognized chaotic models are used to show the performance of the proposed algorithm.
Such hybrid systems always give some new insight into the known issues. The originality is not very high but the proposed model is interesting.
There are two issues worth considering:
- In table 2 it is shown that WNN gives lower values of NMSE than the HAVOK-ML. There is no comment on that fact.
- No empirical data are examined. It would be interesting if any possibly chaotic empirical data are better predicted using the proposed method than using standard ones.
Author Response
1. In Table 2 it is shown that WNN gives lower values of NMSE than the
HAVOK-ML. There is no comment on that fact.
Answer: Thanks for the pertinent review of our manuscript. We have com-
mented this fact on line 112 that
”while the NMSE index shows that the proposed method is second only to the
functional weight WNN state-dependent AR (FWWNN-AR) model”.
2. No empirical data are examined. It would be interesting if any possibly chaotic empirical data are better predicted using the proposed method than using standard ones.
Answer: Thanks for the detailed and valuable advice of the reviewer. Three different type data sets: Lorenz63 system, delay differential Mackey-Glass system and Sunspot serise are applied to verify the accuracy of the proposed HAVOK-ML method. We plan to examined more empirical data such as ocean buoy data, surface wave observation data and so on in our next work. In this paper we only use the data generated from those toy models to compare with other ANN models.

Reviewer 2 Report
This is an interesting method for predicting chaotic time series. The paper is easy to read (except for some minor details). The explanations are lecturing and pictorial - I would especially like to praise Figure 2, which perfectly explains the architecture of one procedural step. Just as if it were on a school board.
The first minor detail that obstructs reading from being perfectly smooth concerns the language. Some sentences are grammatically correct, but they evoke the feeling that they should be expressed in some other way. For example “many studies using machine learning methods to predict chaotic time series have spent a lot of time on how to design the model structure and the setting of model parameters because of the complexity of the learning model.”And sometimes there are real mistakes, e.g. "developed for improve", or "The chosen of the machine learning method is diversity,..." I suggest checking the text by an English expert.
Table 1 mentions the Sunspot and Mackey-Glass time series before they are mentioned or explained. It would be better to start section 3 with an announcement of which chaotic systems will be used to test the proposed method. Then the reader would know the motivation underlying Table 1.
Quite frequently a blank between the words is missing, e.g. “perceptron(MLP) [9]”, “model,a hybrid”, “[24]to decompose” … Sometimes there is an extra blank, e.g. “coefficients [17] . In addition”. There are many examples, so check the text again properly. Consider also this errata: “Min Gan .et. presents”.
“Bold” should not start with a capital letter if it is not at the beginning of the sentence. After an equation, the “where” in the next line should not be indented.
The adjective "bold" should not start with a capital letter if it is not at the beginning of the sentence. After the equation, "where" in the next line should not be indented.
Considering the care with which the figures and tables were created, the spelling mistakes within them are amazing: "sunpot" in Table 1, and, probably owing to cut-and-paste precedures, "actural" appears in all graphs except 4, A2, and A6.
Author Response
- The first minor detail that obstructs reading from being perfectly smooth
concerns the language. Some sentences are grammatically correct, but they evoke the feeling that they should be expressed in some other way. For example “many studies using machine learning methods to predict chaotic time series have spent a lot of time on how to design the model structure and the setting of model parameters because of the complexity of the learning model.”And sometimes there are real mistakes, e.g. ”developed for improve”, or ”The chosen of the machine learning method is diversity,...” I suggest checking the text by an English expert. Answer: Thanks for the pertinent review of our manuscript. We have reexamined the papers for several times and we find the following typos, grammar errors and redundant expressions:
Page1:
Abstract 3rd line: The main idea of HAVOK-ML method is to reconstruct a close linear model to simulate the dynamics of the time series to achieve the purpose of prediction. −→
HAVOK-ML simulates the time series by reconstructing a closed linear model, so as to achieve the purpose of prediction.
Page2:
43th line:developed for improve −→ developed for improving.
45th line:As mentioned above, many studies using machine learning methods to predict chaotic time series have spent a lot of time on how to design the model structure and the setting of model parameters because of the complexity of the learning model. −→
As mentioned above, model structure and parameter tuning are important factors for chaotic time series prediction with machine learning, and a lot of research has focused on it.
Page4:
82th line: a −→ σ
Page5:
105th line: The error increases with increasing predict steps −→ The error in-creases with predict steps.
109th line: At step 10, the obtained RMSE is 9.003e-3, which is significantly better than the result of the literature (0.014 in Figure 4).−→
At step 10, the obtained RMSE is 9.003e-3(Figure 6), which is significantly better than the result of the literature (0.014).
Page8:
126th line:v r −→ v r .
Page9:
142th line: Vr −→ v r . Page10:
154th line:The chosen of the machine learning method is diversity, depending on the property of the external forcing term. −→
It is worth noting that the machine learning method used in HAVOK-ML will vary depending on the property of the external forcing term.
156th line: The experiment results in this paper show that simple machine learning model can achieve high prediction accuracy. −→
The experimental results show that the our method can produce accurate forecasts even with simple machine learning algorithms. - Table 1 mentions the Sunspot and Mackey-Glass time series before they
are mentioned or explained. It would be better to start section 3 with an announcement of which chaotic systems will be used to test the proposed method. Then the reader would know the motivation underlying Table 1. Answer: Thanks for the detailed and valuable advice of the reviewer. We
have made a modification based on this proposal. We add a subsection named ”Introduction” on page 4, and introduced the three series in it. - Quite frequently a blank between the words is missing, e.g. “percep-
tron(MLP) [9]”, “model,a hybrid”, “[24]to decompose” ... Sometimes there
is an extra blank, e.g. “coefficients [17] . In addition”. There are many exam-
ples, so check the text again properly. Consider also this errata: “Min Gan
.et. presents”. Answer: We have revised the paper as requested. - The adjective ”bold” should not start with a capital letter if it is not at the
beginning of the sentence. After the equation, ”where” in the next line should not be indented. Answer: We have modified the paper as requested. In paper ”Where x(t) ∈ R n is the state of ” is changed to ”where x(t) ∈ R n is the state of .” - Considering the care with which the figures and tables were created, the spelling mistakes within them are amazing: "sunpot" in Table 1, and, probably owing to cut-and-paste precedures, "actural" appears in all graphs except 4, A2, and A6. Answer: We have modified the paper as requested.We have changed "sunpot" in Table 1 to "sunspot" and revised all the word "actural" to "actual".
